# Genome-Wide Association Study of Muscle Glycogen in Jingxing Yellow Chicken

**DOI:** 10.3390/genes11050497

**Published:** 2020-04-30

**Authors:** Xiaojing Liu, Lu Liu, Jie Wang, Huanxian Cui, Huanhuan Chu, Huijuan Bi, Guiping Zhao, Jie Wen

**Affiliations:** 1State Key Laboratory of Animal Nutrition, Institute of Animal Sciences, Chinese Academy of Agricultural Sciences, Beijing 100193, China; 18810610203@163.com (X.L.); liulu_0907@126.com (L.L.); wangjie4007@126.com (J.W.); cuihuanxian78@126.com (H.C.); zhaoguiping@caas.cn (G.Z.); 2Yantai Dadi Animal Husbrandry Co., Ltd., Yantai 1265100, China; 15275210115@126.com (H.C.); bihuijuanmail@163.com (H.B)

**Keywords:** muscle glycogen, variation, genome-wide association study, genetic basis, chicken

## Abstract

Glucose metabolism plays an important role in many normal and pathological physiological processes in the body. The breakdown and synthesis of muscle glycogen provides ATP for muscle activities. A genome-wide association study for muscle glycogen was performed in 474 Jingxing yellow chickens to identify significant single nucleotide polymorphisms (SNPs) and insertions and deletions (INDELs) involved in muscle glycogen metabolism. A total of nine SNPs (*p* < 1/699341) and three INDELs (*p* < 1/755733) reached a significant level of potential association. The following results were obtained through a series of analyses, including additive effects and gene function annotation. Two significant SNPs were found in introns 12 and 13 of copine 4 (*CPNE4*) on chromosome 2. The wild-type and mutant individuals had significant differences in glycogen metabolism at two loci (*p* < 0.01 for both). Individuals carrying two mutations had increased muscle glycogen content. According to the gene annotation of chromosome 11, there is a significant INDEL in intron 6 of naked cuticle homolog 1 (*NKD1*). After the INDEL mutation, the glycogen content increased significantly. There was a significant difference between wild-type and mutant individuals (*p* < 0.01). These mutations likely affecting two genes (*CPNE4* and *NKD1*) may affect glycogen storage in a pleiotropic manner. Gene annotation indicates that *CPNE4* and *NKD1* may affect the process of glucose metabolism. Our findings contribute to understanding the genetic regulation of muscle glycogen metabolism and provide theoretical support.

## 1. Introduction

Muscle glycogen is the main source of glucose for muscle glycolysis. Glycolysis is the most primitive metabolic pathway for producing energy in living organisms. Study of muscle glycogen function and influencing factors will reveal the genetic mechanism of glucose metabolism and lead to its application in future molecular breeding research.

Glycogen is the main form of glucose storage in the body, mainly in the form of liver and muscle glycogen. Glycogen is a branched polysaccharide found in the form of granules in the cytosol of the cytoplasm, and is connected by many glucosides. The blood glucose concentration is maintained at a relatively constant level through the synthesis and breakdown of liver glycogen. Numerous studies have suggested that the main function of muscle glycogen synthesis and breakdown is to provide the muscle with energy in the form of ATP. However, under some special physiological conditions, such as intense exercise, muscle glycogen plays an important role in maintaining blood sugar stability and regulating the balance of glucose metabolism [1]. In the case of increased blood glucose levels, the stimulation of insulin secretion causes a rapid increase in the amount of GLUT4, which accelerates the synthesis of muscle glycogen. Conversely, lower blood glucose levels reduce glucose intake and promote the gluconeogenesis process [2]. The glycolytic potential (GP) refers to the total amount of substrate (such as muscle glycogen, glucose-6-phosphate, and glucose) that can be converted into lactic acid in animals or in muscles after slaughter, and the lactate content [3]. Among these compounds, glucose-6-phosphate, glucose, and lactic acid are intermediate products of glycogenesis, in which muscle glycogen plays an important role. Relevant studies have confirmed the relationship between GP and acidic pH [4,5,6].

Glycolysis not only plays an important role in numerous pathological physiological processes in the body, but also has an impact on the flavor quality of meat. It also provides useful molecular markers for the breeding of high-quality broiler chicken breeds. Genetic research on poultry meat quality traits started late. Two studies conducted under the same experimental conditions and on the same broiler production line have shown that the pH value, color, and water retention capacity of meat have moderately high levels of heritability (from 0.35 to 0.57) [7,8]. In addition, meat quality studies performed in turkeys revealed that meat quality traits show intermediate heritability values (from 0.12 to 0.22) [9]. Another study reported that the level of glycogen stored in breast muscle showed high heritability according to the GP during slaughter (*h*^2^ = 0.43), as well as a very strong negative genetic correlation (rg) with pH (rg –0.97) of the final meat, indicating a common genetic control of GP and pH value [10]. A study including breeding experiments using GP as a biological marker to improve meat quality, demonstrated, at least in part, the possibility of applying the intermediate products of glycogen metabolism in pig breeding [11]. 

Whole-genome sequencing is a rapid and sensitive technology used to not only identify single nucleotide polymorphisms (SNPs) and candidate genes, but also shorten the cycle of molecular breeding [12,13]. 

Research on multiple traits in multiple species using genome-wide association studies (GWAS) is being conducted [14,15]. The GWAS of glycogen GP and its related components, namely glycogen, glucose, glucose-6-phosphate, and lactic acid, together with the sequencing analysis results showed a deletion mutation in the splice acceptor site of intron 9 of *PHKG*, an open reading frame change generating a premature stop codon, which led to a 43% increase in GP and a 20% reduction in the moisture of pork [16]. The above genetic results collectively indicate that glycogen content estimated from GP is under close genetic control [10]. In poultry and pigs, both traits can be modified by selection.

GWAS is widely used to analyze the genetic structure of growth, reproduction, and meat quality traits of various poultry. There are no reports about association analyses of poultry glycogen traits, and the analysis data coverage of most other species using microarrays is limited. In this study, we performed GWAS analysis of common SNPs and INDELs by whole-genome sequencing of a chicken line to obtain candidate genes that significantly affect the glycolysis process. It is anticipated to apply the identified molecular markers for the molecular breeding of poultry.

## 2. Materials and Methods 

### 2.1. Ethics Statement

The work was approved by the Animal Management Committee of the Institute of Animal Sciences, Chinese Academy of Agricultural Sciences (IAS-CAAS, Beijing, China). Ethical approval for this experiment was obtained from the IAS-CAAS Animal Ethics Committee (approval number: IAS2019-21).

### 2.2. Resource Population

The experimental animals were Jingxing yellow chicken. Experimental animals were selected for the laboratory-specific intramuscular fat selection line, which had been bred for 16 generations. A total of 474 chickens were selected for this trial, of which 217 were chickens in the intramuscular fat selection line and 256 chickens in the control line (the natural population without selection). All individuals were 98 days of age and allowed to eat and drink ad libitum. Blood samples were collected the day before slaughter by standard venipuncture into a tube with anticoagulant and stored at –20 °C for subsequent genomic DNA extraction. The breast muscle tissue samples from each chicken were stored at –80 °C for subsequent measurements of glycogen content.

### 2.3. Phenotypic Measurement 

A sample of 0.1 g of pectoral muscle tissue was used for measurement. Glycogen is stable in concentrated alkali solutions. The tissue was placed into 300 µL concentrated alkali solution for 20 min before color development to destroy other components and preserve glycogen. Glycogen was then dehydrated with concentrated sulfuric acid to form an aldehyde derivative, which was reacted with anthrone to form a blue compound, which was colorimetrically quantified using a standard glucose solution treated in the same way, and the glycogen content was ultimately calculated. 

### 2.4. Quality Control and Imputation 

Blood collected from veins of the entire population was extracted using the standard phenol/chloroform extraction method. The quantity and quality of the DNA extracted from whole blood samples were determined using a Nanodrop ND-1000 Spectrophotometer (Thermo Fisher Scientific Inc., Waltham, MA, USA) to measure their concentration and integrity, as well as by agarose gel electrophoresis to visually assess DNA integrity. Samples that passed the quality test were sent to the company for 10G whole-genome sequencing. A total of 474 adult females Jingxing yellow chickens (14 weeks of age) were selected to perform the GWAS. Data were first filled with the Beagle v5.0 software [17,18] for all SNP loci, and the filled SNP locus was quality-controlled with the Plink v1.9 software [19]. The standard procedure was to remove sites with a minimum allele frequency (MAF) of less than 5% (MAF < 0.05) and sites with a deletion rate greater than 5% (geno > 0.05). 

### 2.5. Whole-Genome Resequencing

After the examinations, paired-end libraries were generated for each eligible sample using standard procedures. The average insert size was 300–500 bp, and the average read length was PE150 bp (150 bp double-ended). All libraries were separately sequenced on a Novaseq 6000 sequencer (Illumina, San Diego, CA, USA) to an average raw read sequence coverage of ×10 for the natural populations. Accession data code is CRA002643 at https://bigd.big.ac.cn/gsa/.

The filtered pure reads were compared with the reference genome (RefSeq GCF_000002315.4) using the MEM mode of the BWA software (version 0.7.12) (http://bio-bwa.sourceforge.net), and then Picard-tools(version 1.119) (https://broadinstitute.github.io/picard/) and Sam-tools (version 1.9) (http://samtools.sourceforge.net/) were used to obtain the sorted BAM files. Before detecting mutations, the basic quality score recalibration (BQSR) had to be recalibrated. The BQSR involves two steps. In the first step, Base Recalibrator, the Picard tool and stool were used to obtain the classified BAM file for analysis. In the second step, Apply BQSR, the calibration table file obtained in the first step was used to readjust the basic quality value in the original BAM file; this new quality value was then re-exported to a new BAM file. GATK (version 4.0.2.1) selects variants and GATK variants for filtering [20]. The hard filter standard for SNPs is QualByDepth (QD) < 2.0, RMSMappingQuality (MQ) < 40, FisherStrand (FS) > 60.0, StrandOddsRatio (SOR) > 3.0, MappingQualityRankSumTest (MQNankSum) < –12.5; the INDELs hard filter standard is QD < 2.0, FS > 200, SOR > 10.0, MQNankSum < –12.5.

### 2.6. Population Structure and Association Analysis 

As an effective means of GWAS analysis, the genome-wide efficient mixed-model association (GEMMA) v0.94 software, which can perform genomic localization of traits based on population SNP or INDEL information using an efficient exact mixed model approach, was implemented with the valid individuals and SNPs or insertions and deletions (INDELs) for univariate analysis. The method makes exact GWAS computationally feasible for large sample sizes. Independent SNPs or INDELs were used to compute the centered relatedness matrix, and the significance *p*-value level between SNPs or INDELs and phenotypes was calculated from a derived Wald test. The analysis was performed using the GEMMA software [21], and the calculation model was:(1)y=Wα+Xβ+Zu+ε,
where $u\;\sim \;MVN_{m}\left(0,\;\lambda \tau^{-1}K\right)\;$,$\varepsilon \;\sim \;MVN_{n}\left(0,\;\tau^{-1}I_{n}\right)$ n is the number of individuals, m is the number of packets of random effects, y is the n × 1 dimensional quantitative trait phenotype value, W is the covariance matrix of n × c, α is the c × 1 covariate, β is the effective SNP or INDEL, Z is the n × m load matrix, u is m × 1 random effect vector, ε is an n × 1 error vector. τ ^-1^ is the residual variance, λ is the ratio of the variance components, K is the m × m affinity matrix, and I_n_ is the n × n unit matrix.

The genomic inflation factor was calculated with R (https://www.r-project.org/). In this analysis, the first and second principal components were used as covariates, and the affinity matrix (plink, prune, *r*^2^ > 0.2) was established by using unlinked SNPs or INDELs. Individual SNPs or INDELs were solved one by one. Additionally, the significance of the SNP or INDEL was determined by the likelihood ratio test (LRT). 

### 2.7. Gene Identification and Annotation 

The annotated genes that were the nearest or harboring significant SNPs were identified as candidate genes in which significant loci were located [22]. Genes in a specific genomic region [23,24] were detected by using Variant Effect Predictor (VEP) based on the galGal5 assembly supported by Ensembl (oct2018.archive.ensembl.org/Gallus_gallus/Info/Index) accessed in May 2019. 

## 3. Results

### 3.1. Phenotypic Data and Structural Analysis

The maximum and minimum glycogen traits of the 474 individuals were 8.92 mg/g and 0.23 mg/g, respectively, and the average was 2.37 mg/g, with a coefficient of variation of 56%. In the GWAS analysis, population stratification effects may lead to false positive results [25]. As our population was composed of two groups, we calculated the genome-controlled inflation factor (λ) of glycogen to be 1.155. Since this value should ideally be equal to 1, this result indicated that there was a significant population stratification effect. Therefore, in the GWAS, we took the first and second principal components of the population as fixed effects and analyzed them as a whole.

### 3.2. Genome-Wide Association Studies (GWAS) for Single Nucleotide Polymorphism (SNP) and Additive Effect

In this study, for the Q-Q plot presented as Figure 1. The Manhattan map of muscle glycogen is shown in Figure 2. *p*-value correction was first performed using the PLINKv1.9 software [19] for block analysis. This experiment used the Bonferroni correction method to correct the *p*-value. The D’ was set to 0.8, the maximum chain length was 500 Kb, and a total of 699,341 independent linkage disequilibrium (LD) blocks were obtained. 

In the GWAS analysis of muscle glycogen, nine SNPs reached a significant level of potential association (Figure 2), annotated as a 100-Kb gene near the locus (Table 1). There is no gene in the 100 Kb region of chromosome 1 plus or minus, and two SNP sites on chromosome 2 are in the intronic region of the *CPNE4* gene. The chromosome 3 rs13720604 has three genes in this region. The mutation of *TTC7A* mainly acts on the gastrointestinal tract. The TTC7A protein is part of a complex that localizes phosphatidylinositol 4-kinase (PI4K) to the plasma membranes. *SOCS5* is a cytokine signaling inhibitor that acts on IGF-1 receptor signaling and EGF signaling. There are three SNPs on chromosome 4 that reached a significant level. *GRIA3* is a glutamate receptor that acts as a ligand-gated ion channel in the central nervous system and plays an important role in excitatory synaptic transmission. *MFAP3L* protein may participate in the nuclear signaling of EGFR and MAPK1/ERK2 pathway. Chloride voltage-gated channel 3 *(CLCN3*) is a protein coding gene which is associated with cystic fibrosis. Among its related pathways are the activation of cAMP-dependent PKA and the transport of glucose and other sugars, bile salts and organic acids, metal ions, and amine compounds. There are two genes associated with the chromosome 11 locus. *CYLD* encodes a cytoplasmic protein with three conserved cytoskeletal-associated protein-glycine (CAP-GLY) domains that functions as a deubiquitinating enzyme. Gene Ontology (GO) annotations associated with the other gene on this locus, *SNX20,* include phosphatidylinositol binding.

Based on the SNP sites typing results of the whole genome sequencing data, the average genotype frequency and phenotype of nine SNPs were calculated (Table 2). The data in Table 3 reveal that the dominant genotype rs314624646 is TT; the dominant genotypes rs313265900 and rs740265511 are CC; the dominant genotype rs13720604, rs733544657, and rs734443 are GG; and the dominant genotypes rs13720604, rs741487544, and rs315075611 are AA.

The additive effect of alleles is the average of the two homozygous dispersions. According to the cumulative effect, the glycogen content of the rs314624646, rs13720604, and rs740265511 mutations decreased, and the glycogen content of the remaining six SNPs increased. In stepwise conditional analysis, important SNP genotypes were added to the univariate model to elucidate independent signals, and significant differences were found between the nine SNP wild-type and mutant individuals (Appendix A).

### 3.3. GWAS for Insertions and Deletions (INDEL) and Additive Effect

In this study, for the Q-Q plot presented as Figure 3. The results of the GWAS analysis indicated that there were three significant levels of metabolism related INDELs in the whole genome (*p* < 1.32 × 10^−6^) (Figure 4), which were located on chromosomes 1, 3, and 11 (Table 4). Genes such as *NKD1* and *FOSL2* were identified. The INDEL of chromosome 11 is located in the *NKD1* gene, which encodes a protein that is an inhibitor of the canonical WNT signaling pathway. Glycogen synthase kinase 3A (GSK3A) is among the activated target genes of NKD1. At 29.7 Kb upstream of chromosome 3, 27425548 is the *FOSL2* gene, which, as a dimer with JUN, activates LIF transcription and *CEBPB* transcription in PGE2-activated osteoblasts. *CEBPB* plays a major role in adipogenesis and gluconeogenesis. There is no gene in the region of chromosome 1 7169549 plus or minus 100 Kb.

The same method as that used for the calculation of SNPs in Section 3.2 was used to calculate the average genotype frequency and phenotype of INDELs (Table 3). The results indicate that the three INDELs (sorted from small to large on the chromosome) dominant genotypes are wild-type, heterozygous, and wild-type.

The same additive effect found that the three INDELs resulted in varying degrees of glycogen content. In stepwise conditional analysis, important INDEL types were added to the univariate model to identify independent signals, and significant differences were found between the three INDELs wild-type and mutant individuals (Appendix A).

## 4. Discussion

Muscle glycogen and glucose in the glucose metabolism cycle are the main compounds that maintain normal metabolism and function in skeletal muscle cells. Glycogen also affects meat quality through its GP [26]. 

The Efficient Mixed-Model Association eXpedited (EMMAX) test and the Genome Association and Prediction Integrated Tool (GAPIT) both adopt the strategy of unchanged variance components in the fixed zero model to improve the operation speed. This is actually an approximate algorithm which is not as accurate as GEMMA. GEMMA can directly use the plink binary format data without complex data format conversion. It is more comprehensive and can perform single-label GWAS, multilabel GWAS and multitrait GWAS analysis.

INDEL marker refers to the insertion or deletion of nucleotide fragments of different sizes at the same site in the genome between different individuals of the same species or between different individuals of different species [27]. The distribution of INDEL markers in the genome is second only to SNP markers, but it has good stability, high polymorphism, and simple banding [28]. We performed GWAS for SNPs and INDELs associated with glycogen metabolism, both of which found significant sites that enriched the results, and identified more candidate SNP or INDEL sites and genes [21].

This trial aims to identify relevant SNP or INDEL sites and genes that affect chicken muscle glycogen content. Although many studies have attempted to identify the genetic determinants of chicken meat traits, most of them focused on the use of chips to study the visual traits of the slaughtered animal, and to date, there has been no related research on glucose metabolism indicators, such as muscle glycogen.

According to the association analysis of data binding phenotypes of the 474 individuals using whole-genome sequencing, two SNP sites on chromosome 2 are on the intronic region of the gene *CPNE4*, which belongs to the highly-conserved copine family. It encodes a calcium-dependent, phospholipid-binding protein which is significantly expressed in cancer tissues [29]. To date, there have been no reports of *CPNE4* affecting glycogen content. The two SNP mutations in chromosome 2 have a glycogen-increasing effect; the SNP is located on the intronic region of the gene, and may have an enhancing effect by regulating the promoter. However, there are fewer individuals with mutations, which may be due to the small size of the population sample, and a larger sample and further research are needed in the future. *CPNE4* can be considered as a candidate gene that affects muscle glycogen of chicken muscle, and provides a reference for marker-assisted selection of Jingxing yellow chicken. However, the related mechanism needs further verification.

The site at 62.1 Kb upstream of the rs733544036 on chromosome 4 is located adjacent to the glutamate ionotropic receptor AMPA type subunit 3 (*GRIA3*) gene. It has been reported that after astrocyte uptake of glutamate, the glutamate transporter initiates the glycolysis process [30,31,32], *GRIA3* is involved in muscle glycogen synthesis and affects muscle glycogen content and changes in its content. The additive effect showed that the phenotype increased after mutation at this site; the mean values of the phenotypes of the wild-type and mutant individuals were significantly different. At 32.8 Kb upstream of the rs315075611 is the gene *CLCN3*, which encodes a member of the voltage-gated chloride channel (ClC) family. This ClC protein is required for lysophosphatidic acid (LPA)-activated Cl-current activity and fibroblast-myofibroblast differentiation. Indeed, the identification of receptors in lysophospholipid cells revealed that LPA is a transcellular PPAR gamma (PPARG) agonist [33], and PPARG regulates two enzymes involved in glucose metabolism by inhibiting TGFB1-induced mitochondrial activation [34]. As an inducer of brown fat, PPARG plays an important role in regulating glucose oxidation and glucose uptake in brown fat [35].

A total of three INDELs in the GWAS analysis reached a potential level of association. The INDEL with the most significant association to the glycogen metabolism is the locus in the *NKD1* gene on chromosome 11. *NKD1* is an inhibitor of the canonical WNT signaling pathway and is involved in β-catenin-dependent WNT signaling. *GSK3A* is one of the major target genes for activation of the WNT pathway. *GSK3A* was originally identified as one of five phosphorylating [36,37], rate-limiting glycogen-synthase kinases that help insulin regulate glycogen synthesis by phosphorylating and inhibiting glycogen synthase 1 (GYS1) activity, and thereby inhibiting glycogen synthesis [38,39]. 

Close to the chromosome three INDEL is the gene *FOSL2*, whose protein product activates *CEBPB* transcription in PGE2-activates osteoblasts. It is rapidly expressed during adipogenesis, induces *CEBPA* and *PPARG* after activation by phosphorylation, and activates a series of adipocyte genes that cause the adipocyte phenotype. Glucose and lipid metabolism are a research hotspot. A negative correlation between muscle GP and its fat content (*r* = –0.27, *p* < 0.05) has been reported [40]. We speculate that the INDEL affects *FOSL2*, thereby influencing the expression of lipid metabolism genes, and ultimately, the glycogen content.

A total of nine SNPs and three INDELs reached a significant level, and the SNP and INDEL genotypes were added to the univariate model to identify independent signals in a stepwise conditional analysis, which revealed nine SNPs and three INDELs wild type and mutations. Significant differences existed between individuals, and both site mutations and insertional deletions resulted in significant changes in glycogen content. Additive effects and other analyses showed that the loci identified by our analysis are valid and can be used for subsequent further verification, and also provide a relevant theoretical basis for related research.

In view of the described GWAS analysis, the relevant loci, and the identification of genes near each locus, we speculate that *CPNE4* and *NKD1* may affect the muscle glycogen metabolism process in chickens.

## 5. Conclusions

This study comprised a GWAS analysis of SNP and INDEL associated with chicken glycogen traits, revealing genes associated with muscle glycogen content. The identified glycogen content-associated genes, *CPNE4* and *NKD1*, may be important candidate genes involved in the regulation of muscle glycogen content by influencing glucose metabolism. Our findings provide new insights into the genetic mechanism of muscle glycogen metabolism.

## Figures and Tables

**Figure 1 genes-11-00497-f001:**
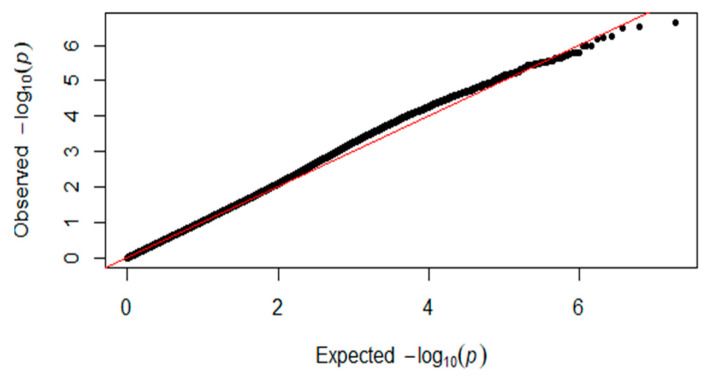
Quantile–quantile plots of *p*-values for muscle glycogen. The x-axis is the expected −log10 *p*-value and the y-axis is the observed −log10 *p*-value.

**Figure 2 genes-11-00497-f002:**
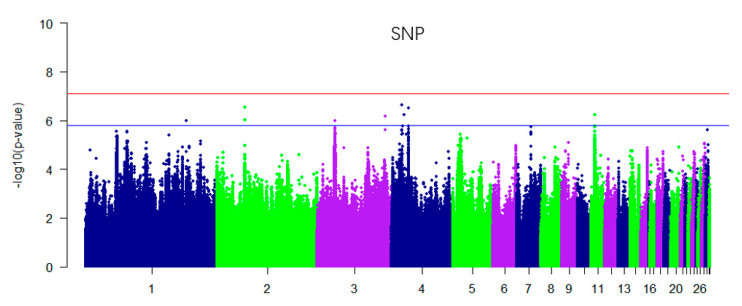
Manhattan plot derived from a genome-wide association studies (GWAS) single nucleotide polymorphisms (SNPs) at muscle glycogen. Each dot corresponds to a SNP within the dataset, while the horizontal blue and red lines denote the genome-wide significance (7.15 × 10^−8^) and suggestive significance thresholds (1.43 × 10^−6^), respectively. The Manhattan plot contains –log_10_ observed *p*-values for genome-wide SNPs (y-axis) plotted against their corresponding position on each chromosome (x-axis).

**Figure 3 genes-11-00497-f003:**
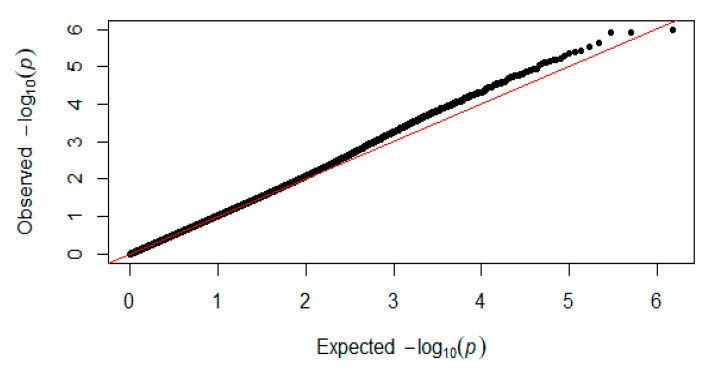
Quantile–quantile plots of *p*-values for muscle glycogen. The x-axis is the expected −log_10_
*p*-value and the y-axis is the observed −log_10_
*p*-value.

**Figure 4 genes-11-00497-f004:**
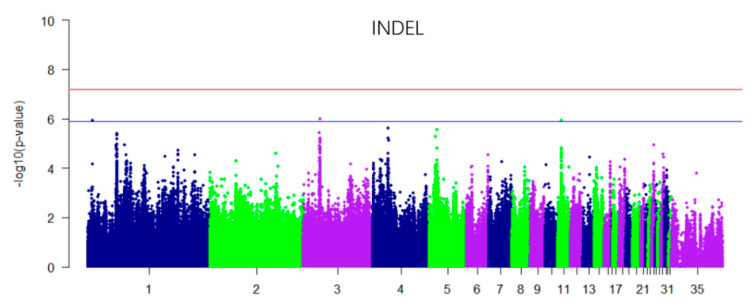
Manhattan plot derived from a GWAS (INDEL) at muscle glycogen. Each dot corresponds to an INDEL within the dataset, while the horizontal blue lines denote the suggestive significance thresholds (1.32 × 10^−6^). The Manhattan plot contains –log_10_ observed *p*-values for genome-wide SNPs (y-axis) plotted against their corresponding position on each chromosome (x-axis).

**Table 1 genes-11-00497-t001:** SNPs with genome-wide significance for muscle glycogen.

Chromosome	SNP ID	Position (BP)	p_Wald	Nearest Gene	Location (Kb)
1	rs314624646	150449184	1.05 × 10^−06^	/	/
2	rs313265900	42074647	2.89 × 10^−07^	*CPNE4*	Intronic
2	NEW	42078810	9.95 × 10^−07^	*CPNE4*	Intronic
3	rs13720604	26983490	1.04 × 10^−06^	*MCFD2*	D12.4
				*TTC7A*	U26.6
				*SOCS5*	D72.7
3	rs740265511	101450476	6.58 × 10^−07^	*OSR1*	D18.5
4	rs733544036	15746019	2.29 × 10^−07^	*GRIA3*	U62.1
4	rs741487544	18209706	5.66 × 10^−07^	*IDS*	U53.5
4	rs315075611	25507719	3.14 × 10^−07^	*MFAP3L*	D30.8
				*CLCN3*	U32.8
				*HPF1*	D22.1
11	rs734443657	6224928	6.02 × 10^−07^	*CYLD*	D50.2
				*SNX20*	U73.2

U: Upstream, D: Downstream, *CPNE4: Copine 4, MCFD2: Multiple Coagulation Factor Deficiency 2, TTC7A: Tetratricopeptide Repeat Domain 7, SOCS5: Suppressor Of Cytokine Signaling 5A, OSR1: Odd-Skipped Related Transcription Factor 1, GRIA3: Glutamate Ionotropic Receptor AMPA Type Subunit 3, IDS: Iduronate 2-Sulfatase, MFAP3L: Microfibril Associated Protein 3 Like, CLCN3: Chloride Voltage-Gated Channel 3, HPF1: Histone PARylation Factor 1, CYLD: CYLD Lysine 63 Deubiquitinase, SNX20: Sorting Nexin 20*.

**Table 2 genes-11-00497-t002:** Allelic frequency, phenotypic mean, additive effect of significant SNP.

Chr	Position (BP)	ref	alt	freq_ref|ref	freq_ref|alt/alt|ref	freq_alt|alt	pheno_ref|ref	pheno_ref|alt	pheno_alt|alt	Additive Effect
1	150449184	C	T	0.1308	0.3059	0.5633	3.1108	2.4512	2.1559	–0.4775
2	42074647	C	T	0.8376	0.1414	0.0211	2.2755	2.5604	4.8972	1.3108
2	42078810	A	C	0.8586	0.1224	0.0190	2.2726	2.7663	4.2772	1.0023
3	26983490	G	A	0.4515	0.4051	0.1435	2.6424	2.2138	1.9615	–0.3405
3	101450476	C	T	0.5380	0.2025	0.2595	2.6404	2.1277	2.0027	–0.3188
4	15746019	G	T	0.8658	0.1216	0.0126	2.2689	2.8665	4.5671	1.1491
4	18209706	A	G	0.8692	0.1097	0.0211	2.2713	2.8023	4.2408	0.9847
4	25507719	A	G	0.8847	0.1048	0.0105	2.2757	2.9680	4.3935	1.0589
11	6224928	G	A	0.6603	0.2869	0.0527	2.1946	2.6683	2.9648	0.3851

ref means wild type; alt means mutant; freq_ref|ref refers to the genotype frequency of a wild-type individual; freq_ref|alt/alt|ref refers to the genotype frequency of heterozygous individuals; freq_alt|alt refers to the genotype frequency of a mutant individual; pheno_ref|ref refers to the mean of the wild-type individual phenotype; pheno_ref|alt refers to the average phenotype of heterozygous individuals; pheno_alt|alt refers to the average phenotype of mutant individuals.

**Table 3 genes-11-00497-t003:** Allelic frequency, phenotypic mean, additive effect of significant INDEL.

Chr	Position (BP)	ref	alt	freq_ref|ref	freq_ref| alt/alt |ref	freq_ alt|alt	pheno_ref|ref	pheno_ref| alt	pheno_ alt|alt	Additive Effect
1	7169549	G	GACAA	0.8421	0.1308	0.0219	2.2696	2.8068	3.9756	0.8530
3	27425548	T	TA	0.3861	0.3991	0.2148	2.1143	2.3646	2.8366	0.3612
11	6376492	AG	A	0.5416	0.3539	0.1045	2.1395	2.5656	2.9383	0.3994

ref means wild type; alt means insertions and deletions; freq_ref|ref refers to the genotype frequency of a wild-type individual; freq_ref|alt/alt|ref refers to the genotype frequency of heterozygous individuals; freq_alt|alt refers to the genotype frequency of a mutant individual; pheno_ref|ref refers to the mean of the wild-type individual phenotype; pheno_ref|alt refers to the average phenotype of heterozygous individuals; pheno_alt|alt refers to the average phenotype of mutant individuals.

**Table 4 genes-11-00497-t004:** INDELs with genome-wide significance for glycogen traits.

Chromosome	Position (BP)	p_Wald	Nearest Gene	Location (Kb)
1	7169549	1.23 × 10^−06^	/	/
3	27425548	1.03 × 10^−06^	*FOSL2*	U29.7Kb
11	6376492	1.18 × 10^−06^	*NKD1*	Intronic

*FOSL2: FOS Like 2, AP-1 Transcription Factor Subunit; NKD1: naked cuticle homolog 1*.

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
