# Peer review of "Genome-Wide Association Study of Muscle Glycogen in Jingxing Yellow Chicken"

_genes, 2020, doi:10.3390/genes11050497_

Round 1

Reviewer 1 Report

The study, based on GWAS analysis of SNPs and INDELs, shows two genes, which could be implicated in muscle glycogen level regulations. The research is original and novel. The results provide an advance in current knowledge.

The author’s statement in the abstract (line 12-16) is not supported in the results and should be mitigated. This conclusion is not supported by the results. There is no research proving that the identified SNPs and INDELs are involved in muscle glycogen metabolism.

The figure's description is not precise and not sufficient. For example, fig 2 and 4 have the same description, although they refer to different results.

The authors measure the glycogen level in pectoral muscle. This should be precisely indicated in the text. For example the terms “glycogen phenotype” (line 150), “glycogen content” (line 173, 174), “3.3. GWAS for glycogen […]” (line 191) is not precise. The other example is the graph title “Glycogen” (Fig 2 and 4), which is not sufficient and precise.

The study is correctly designed.

The conclusions described in “Discussion” are interesting and could have an impact on future research.

The style of the article abstract needs improvement. Authors use a lot of repetitions such as “regulate glucose metabolism” and “affect the process of glucose metabolism” in one sentence (lines 23-25), or “glycogen content” 3 times in lines 18-21.

The research use animals (chickens). The project obtained ethical approval from the dedicated Committee.

Reviewer 2 Report

The manuscript is well written and the results collaborate to improve the knowledgment of the glycogen content in chicken broiler and its effect on meat quality. However, some points need to be revised and better explained.

Line 42: The sentence " hydraulic properties of meat" is not usual. The authors could change it.
Line 59: Replace the comma to end point.

Introduction: The authors cited the previous studies about the same issue, however in pig. The authors should change their introduction citing chicken studies. They also should clarify the importance, reason and the differential of their study.

Why they used Jingxing yellow breed? Explain it.

Line 101: The authors could present the could present the Whole-genome resequencing coverage.

Line 134: How the gene annotation was performed.

Line 179: The quality of the Figures 2 and 4 should be improved.

The pH, color, WHC, tenderness and processing yields were also measured by the authors? These informations ae very important to discuss their results.

Round 2

Reviewer 2 Report

All of my questions and requests are answered. The authors addressed properly their explanations.

Author Response

Thank you for your careful review and comments.